# Iridal-Type Triterpenoids Displaying Human Neutrophil Elastase Inhibition and Anti-Inflammatory Effects from *Belamcanda chinensis*

**DOI:** 10.3390/molecules26216602

**Published:** 2021-10-31

**Authors:** Jeong Ho Kim, Yeong Jun Ban, Aizhamal Baiseitova, Marie Merci Nyiramana, Sang Soo Kang, Dawon Kang, Ki Hun Park

**Affiliations:** 1Division of Applied Life Science (BK21 plus), IALS, Gyeongsang National University, Jinju 52828, Korea; rwg2610@gnu.ac.kr (J.H.K.); banyoung972@naver.com (Y.J.B.); aizhabaiseitova@gmail.com (A.B.); 2Department of Physiology, College of Medicine and Institute of Health Sciences, Gyeongsang National University, Jinju 52727, Korea; mariemerci1994@naver.com; 3Department of Anatomy, Institute of Health Sciences, College of Medicine, Gyeongsang National University, Jinju 52727, Korea; kangss@gnu.ac.kr

**Keywords:** *B. chinensis*, iridal-type triterpenoids, human neutrophil elastase, anti-inflammation

## Abstract

The aim of this study is to explore anti-inflammatory phytochemicals from *B. chinensis* based on the inhibition of pro-inflammatory enzyme, human neutrophil elastase (HNE) and anti-inflammatory activities in lipopolysaccharide (LPS)-stimulated RAW264.7 macrophage. Three stereoisomers of iridal-type triterpenoids (**1**–**3**) were isolated from the roots of *B. chinensis* and their stereochemistries were completely identified by NOESY spectra. These compounds were confirmed as reversible noncompetitive inhibitors against HNE with IC_50_ values of 6.8–27.0 µM. The binding affinity experiment proved that iridal-type triterpenoids had only a single binding site to the HNE enzyme. Among them, isoiridogermanal (**1**) and iridobelamal A (**2**) displayed significant anti-inflammatory effects by suppressing the expressions of pro-inflammatory cytokines, such as iNOS, IL-1β, and TNF-α through the NF-κB pathway in LPS-stimulated RAW264.7 cells. This is the first report that iridal-type triterpenoids are considered responsible phytochemicals for anti-inflammatory effects of *B. chinensis*.

## 1. Introduction

*Belamcanda chinensis* (L.) DC known as *She Gan* has been used in Traditional Chinese Medicine (TCM) for the clinical treatment of respiratory diseases including bronchitis, asthma, sore throat and so on [1], which have been listed in Chinese Pharmacopoeia (Committee for the Pharmacopoeia of P.R. China, 2015). *B. chinensis* belongs to the Iridaceae family and is widely distributed in North America and South-East Asia [2]. The phytochemical constituents of *B. chinensis* are flavonoids including isoflavones, flavones, and flavonols [3]. Many reports elucidated that isoflavones are major metabolites among phenolic compounds and the uniqueness of this plant is to have triterpenoids with iridal skeleton [4]. This study focused on the phytochemicals which could be attributed to the anti-inflammation potential of *B. chinensis*, based on human neutrophil elastase (HNE) inhibition and the alleviation effects of the pro-inflammatory cytokines.

Human neutrophil elastase (HNE, EC 3.4.21.37) is a kind of serine protease and is mostly secreted in neutrophil granules of leukocytes in human blood [5]. This enzyme plays a crucial role in the inherent immune system for host defense against pathogen and inflammation [6,7]. The activity of the HNE enzyme is regulated by endogenous serine protease inhibitors, including α-1 antitrypsin (AAT), also known as alpha-1 proteinase inhibitor (α-1 PI), to prevent unrestrained proteolysis, which leads to immoderate inflammation in the body [8]. The overexpression of HNE by oxidative stress is associated with the increase of the expression of pro-inflammatory cytokines such as interleukin (IL)-1β, IL-8, TNF-α, and nuclear factor-kappa B (NF-κB) [9]. In particular, the activation of NF-κB leads to its sequestration from the cytoplasm and translocation to the nucleus. It induces the transcription of pro-inflammatory cytokines that stimulate the inflammatory processes [10]. The imbalance between HNE and its regulators causes various chronic diseases such as chronic obstructive pulmonary disease (COPD) [11], atherosclerosis [12], arthritis [13], and liver cirrhosis [14] based on inflammation. Recently, there have been many reports suggesting that HNE inhibitors of medicinal plants are related to anti-inflammatory effects [15,16,17]. Certainly, anti-inflammation studies are mainly focused on the phytochemicals which could alleviate the inflammatory symptoms induced by oxidative factors. The lead structures of representative anti-inflammatory compounds, including curcumin, resveratrol, epigallocatechin-3-gallate (EGCG), and quercetin [18], belong to phenolics, but there are a few reports on triterpenoids [19].

This study attempts to disclose the anti-inflammatory triterpenoids in the roots of *B. chinensis* based on pro-inflammatory enzyme (HNE) inhibition and anti-inflammation activities in LPS-stimulated RAW264.7 cells. It included the isolation of phytochemicals, their structural identification, and HNE inhibition. The anti-inflammatory activities of isolated triterpenoids were determined by the suppression of the pro-inflammatory cytokines in LPS-stimulated RAW264.7 cells.

## 2. Results and Discussion

### 2.1. Isolation and Identification of Iridal-Type Triterpenoids

In the activity-guided fractionation of the HNE enzyme, the hexane fraction of the methanol extract showed effective inhibition (78% inhibition, 100 µg/mL). Four iridal-type triterpenoids were purified from the hexane fraction and their structures were determined by their spectroscopic data (including 2D-NOESY results) and comparison with previous studies [20,21,22] (Appendix A). Thus, the isolated triterpenoids (**1**–**3**) were identified as isoiridogermanal (**1**), iridobelamal A (**2**) and iridobelamal B (**3**), respectively (Figure 1). For example, the most active HNE inhibitory compound **3** was obtained as a colorless oil having the molecular formula C_31_H_48_O_5_ and eight degrees of unsaturation, as established by the [M + Na]^+^ ion at 500.3514 (Calcd. 500.3502) in the HRESIMS. The analysis of degrees of unsaturation indicated dicyclic skeleton with six double bonds. A typical characteristic of the iridal-type triterpenoids, a homofarnesyl group at C-11 of **3** was confirmed by successive connectivity from H-11 (δ_H_ 4.88) to H-30 (δ_H_ 1.62) and HMBC correlations between C-11 (δ_C_ 76.2) and H-12 (δ_H_ 5.48). The lower field quaternary carbon (C-6, δ_C_ = 60.7) was confirmed as the common atom of a spiro-structure. The *Z*-configuration of **3** had NOESY relationships of H-1 (δ_H_ 10.24) with H-23 (δ_H_ 1.93), H-24 (δ_H_ 1.30), and H-5 (δ_H_ 3.66). Thus, compound **3** was confirmed as iridobelamal B.

### 2.2. Human Neutrophil Elastase Inhibition

Human neutrophil elastase is controlled by its endogenous inhibitors such as α-1 antitrypsin, but oxidative stress causes these enzymes to reduce their activities on HNE [23]. It is generally believed that the additional activity of HNE is linked to inflammation [13]. All isolated triterpenoids (**1**–**3**) were screened for inhibitory activities against HNE. The inhibitory activity was assayed according to a standard procedure by spectrophotometric method using MeOSuc-AAPV-*p*NA as a substrate. Compounds **1**–**3** exhibited potent and dose-dependent inhibition against HNE with IC_50_ values of 6.8–27.0 µM. In particular, all triterpenoids showed a selective inhibition to HNE in comparison with other hydrolases such as α-glucosidase, PTP1B, and neuraminidase (Figure 2a). This is the first report on the effective inhibitory effect of iridal-type triterpenoids on HNE. The inhibitory activity was affected by subtle changes in structure. The spiro-type compound **3** showed 2–3 fold better inhibition than monocyclic compounds (**1** and **2**) as shown in Figure 2b and Table 1.

In a kinetic study, the enzyme activity was measured at various inhibitor concentrations over a series of substrate concentrations. All compounds manifested a similar relationship between enzyme activity and concentrations. The possible kinetic inhibition modes were obtained through double-reciprocal plots of the Michaelis–Menten equation. The reversibility of compound **3** to HNE enzyme was proved by plotting residual enzyme activities versus enzyme concentrations at different concentrations of compound **3**, resulting a family of straight lines with a common *y*-axis intercept (Figure 2c). As depicted in Figure 3c,f, the inhibition kinetics elucidated by the Lineweaver-Burk plots indicated that compound **3** is a noncompetitive inhibitor. Because increasing the concentration of **3** resulted in a family of lines with a common intercept on the *x*-axis but with different gradients. It illustrated that *V*_max_ decreased without change of *K*_m_ in the presence of an increasing concentration of **3**. A *K*_i_ value of **3** was calculated as 6.2 µM by Dixon plots. Other compounds (**1** and **2**) also displayed noncompetitive inhibition behaviors (Figure 3).

### 2.3. Binding Affinities to Human Neutrophil Elastase

It is rare that triterpenoids have inhibition activities against the HNE enzyme. Thus, HNE inhibitory activities of compounds (**1**–**3**) were doubly confirmed by using the fluorescence (FS) quenching effects based on an enzyme binding affinity. HNE has the intrinsic fluorescence property based on mostly tryptophan residues, Trp-12, Trp-127, and Trp-212 [24]. This fluorescent intensity might be changed by a function of ligand concentration when the enzyme interacts with another ligand. A significant emission was not observed from any of the other components in the assay mixture under the measurement conditions (λ_Ex_ = 260 nm, λ_Em_ = 280–400 nm). Figure 4a–c showed that the dose-dependent FS quenching effect is proportional to the increase in inhibitor concentration. The FS quenching degrees also accorded closely with inhibitory potencies: **3** (IC_50_ = 6.8 µM) vs. **2** (IC_50_ = 27.0 µM).

The Stern–Volmer quenching constant (*K*_sv_), the binding constant (*K*_A_), and the number of binding sites (*n*) were analyzed in Equations (5) and (6) (Experimental section). The *K*_sv_ values were ranked in the following order **3** > **1** > **2**, which were essentially in agreement with the order of the inhibitory potencies (Figure 4d). The values of *K*_A_ increased from 0.0199 up to 0.0743 (×10^6^ L mol^−1^) by inhibitory potencies (Table 2). The values of *n* of all inhibitors (**1**–**3**) were approximately one (0.97~1.00), indicating that a single binding site exists in HNE for iridal-type triterpenoids.

### 2.4. Anti-Inflammatory Effect of Triterpenoid ***1*** and ***2***

The cytotoxic effects of the triterpenoids (**1**–**3**) were measured at various concentrations (0.1, 0.3, 1, 3, and 10 µg/mL) in RAW264.7 cells for 24 h using MTT assays. As shown in Figure 5a, the concentrations tested in this study did not induce cell death except for 10 μg/mL. Triterpenoids **1** and **2** at a concentration of 10 μg/mL reduced cell viability by 28.8 ± 4.4% and 23.4 ± 5.1%, respectively. To confirm the effect of the triterpenoids on the cellular inflammatory condition, 1 and 3 μg/mL concentrations were used in the following experiments.

LPS, a major component of the outer Gram-negative bacteria membrane, was used to create the cellular inflammatory condition, and LPS-treated RAW264.7 cells secrete a variety of inflammatory mediators [25]. LPS-treated RAW264.7 cells undergo morphological changes, such as cell enlargement, flattening, cellular spreading, and vacuole formation, which indicates that RAW264.7 cells have been activated [26].

Among the four triterpenoids, **1** and **2** ameliorated the LPS-induced morphological changes in RAW264.7 cells. In treatment, except with LPS, the triterpenoids did not induce macrophage activation as shown in Figure 5b. In response to LPS, the mRNA expression level of iNOS and NO production in RAW264.7 cells were stimulated markedly, whereas triterpenoids **1** and **2** significantly reduced LPS-induced iNOS and NO upregulation (*n* = 5, *p* < 0.05). Furthermore, triterpenoid **2** showed a dose-dependent inhibition at the concentrations of 1 and 3 μg/mL (*n* = 4, *p* < 0.05, Figure 5d). However, these triterpenoids did not influence the expression of the housekeeping gene GAPDH. Because triterpenoids **1** and **2** were found to inhibit pro-inflammatory mediator NO significantly, we examined their effect on LPS-stimulated pro-inflammatory cytokine signals such as IL-1β and TNF-α. As shown in Figure 6a,b, the upregulated mRNA levels and the concentrations of IL-1β and TNF-α were noticeably decreased by triterpenoids **1** and **2**. (*n* = 6, *p* < 0.05).

The NF-κB activation induced by LPS led to the translocation of the NF-κB signal from the cytoplasm to the nucleus. Along with Bay 11-7085, which is the NF-κB inhibitor, pre-treatment with triterpenoids **1** and **2** suppressed the NF- κB translocation to the nucleus in the immunocytochemical staining (Figure 6c). In particular, triterpenoid **2** inhibited nuclear translocation of p65 protein, the major subunit of NF-κB, in a dose-dependent manner, as shown in Figure 6d (*n* = 3, *p* < 0.05). However, these triterpenoids did not affect the expression of β-actin and lamin, which are housekeeping proteins of the cells. In summary, triterpenoids **1** and **2** effectively suppressed the expression levels of pro-inflammatory signals such as iNOS, IL-1β, and TNF-α by blocking the NF-κB pathway (Figure 6e). Thus, they both had anti-inflammatory potential in LPS-stimulated RAW264.7 cells.

## 3. Materials and Methods

### 3.1. General Experimental Procedures

Open column chromatography was conducted using octadecylsilanized (ODS) silica gel (50 µm, YMC Ltd., Kyoto, Japan). Preparative recycling high pressure liquid chromatography (HPLC) was carried out by LC-9130G NEXT (Jai Co., Ltd., Tokyo, Japan) using AQ C18 (S-10 µm, 12 nm, YMC, Kyoto, Japan) and Acclaim Polar Advantage II C-18 (S-5 µm, 12 nm, Thermo Fisher Scientific, Waltham, MA, USA). ^1^H and ^13^C-NMR, as well as 2D NMR data were recorded by a Bruker AM500 spectrometer (Bruker, Billerica, MA, USA) using Acetone-d6 or Chloroform-d with tetramethylsilane (TMS) as an internal standard. UV spectra were measured using a DU650 spectrophotometer (Beckman Coulter, Brea, CA, USA). HRESIMS were carried out by a Vion (Waters, Milford, MA, USA). Specific rotation ([α]) was estimated using a P-2000 Digital Polarimeter (JASCO, Tokyo, Japan). Enzymatic assays were conducted by a SpectraMax M3 Multi-mode Microplate Reader (Molecular Devices, San Jose, CA, USA). All chemicals for analyses were of first grade. The root of B. chinensis [imported from China with permission of the Korean Food and Drug Administration (KFDA)] was purchased from a Korean pharmaceutical market (Jinju, Korea).

### 3.2. Extraction and Isolation

The dried roots of *B. chinensis* (2.4 kg) were extracted with methanol (10 L × 3) for 2 weeks at room temperature. The accumulated filtrate was evaporated to yield a pale yellow residue (54 g), which was suspended in water (0.5 L) and further fractionated successively with hexane (1L × 3) and ethyl acetate (1L × 3). The hexane fraction (8.1 g) was subjected to ODS silica gel (200 g) using a gradient of water to methanol (5:1 to 1:10, *v*/*v*), which provided five fractions (A–E). HNE inhibitory fractions D–E (1.8 g) were portioned using a preparative HPLC with reversed silica gel CC (250 mm × 30 mm, S-10 µm, 12 nm, YMC) and eluted using a gradient of methanol in water (60% to 100%, *v*/*v*) at a rate of 10 mL/min, to yield thirty subfractions (D1–D30). Subfractions D6–D11 (180 mg) were further purified by recycling HPLC with reversed silica gel CC (250 mm × 30 mm, S-5 µm, 12 nm, Thermo Fisher Scientific, Waltham, MA, USA) using a isocratic elution with H_2_O:ACN (3:7, *v*/*v*) to give rise to compound **1** (7.2 mg) and compound **2** (68.3 mg). Subfractions D24–D27 (95 mg) provided **3** (32.1 mg) through equal recycling HPLC with a reversed silicagel CC using an isocratic elution with H_2_O:ACN (1:9, *v*/*v*).

#### 3.2.1. Isoiridogermanal (**1**)

Colorless oil. HRESIMS [M + Na]^+^ 474.3706 (calcd. for C_30_H_50_O_4_ 474.3709). [α${]}_{D}^{25}$ + 36.7 (c 0.1, EtOH). ^1^H-NMR (500 MHz, Acetone-*d*_6_): δ 1.10 (3H, s, H-26), 1.16 (3H, s, H-27), 1.18 (1H, m, H-10a), 1.26 (1H, m, H-24a), 1.32 (1H, m, H-10b), 1.40 (1H, m, H-24b), 1.55 (3H, s, H-28), 1.60 (3H, s, H-30), 1.62 (3H, s, H-29), 1.65 (1H, m, H-8a), 1.68 (3H, s, H-22), 1.79 (1H, m, H-23a), 1.83 (3H, s, H-3), 1.86 (1H, m, H-11a), 1.87 (1H, m, H-8b), 1.96 (1H, m, H-11b), 2.02 (2H, m, *J* = 6.5 Hz, H-18), 2.03 (1H, m, H-23b), 2.07 (2H, m, H-19), 2.22 (2H, m, H-15), 2.55 (1H, br t, *J* = 13.8 Hz, H-9a), 2.60 (1H, td, *J* = 13.8, 4.7 Hz, H-9b), 3.31 (1H, br d, *J* = 11.1 Hz, H-5), 3.61 (2H, t, *J* = 6.4 Hz, H-25), 3.92 (1H, dd, *J* = 7.8, 5.1 Hz, H-14), 5.06 (1H, m, H-20), 5.07 (1H, m, H-16), 5.25 (1H, t, *J* = 7.0 Hz, H-12), 10.18 (1H, s, H-1) (see Appendix A).

#### 3.2.2. Iridobelamal A (**2**)

Colorless oil. HRESIMS [M + Na]^+^ 474.3739 (calcd. for C_30_H_50_O_4_ 474.3709). [α${]}_{D}^{25}$ + 42.0 (c 0.01, EtOH). ^1^H-NMR (500 MHz, Acetone-*d*_6_): δ 1.09 (3H, s, H-26), 1.13–1.32 (2H, m, H-10), 1.16 (3H, s, H-27), 1.33–1.41 (2H, m, H-24), 1.60 (3H, s, H-28), 1.60 (3H, s, H-30), 1.62–1.71 (1H, m, H-8a), 1.63 (3H, s, H-29), 1.68 (3H, s, H-22), 1.79 (1H, m, H-8b), 1.80 (1H, m, H-23a), 1.80 (3H, s, H-3), 1.90–1.96 (1H, m, H-11a), 1.99 (1H, m, H-11b), 2.02 (2H, m, H-18), 2.07 (2H, m, H-19), 2.10 (1H, m, H-23b), 2.13–2.33 (2H, m, H-15), 2.59 (1H, br t, *J* = 12.5 Hz, H-9a), 2.79 (1H, br d, *J* = 9.9 Hz, H-5), 3.22 (1H, br d, *J* = 13.9 Hz, H-9b), 3.61 (2H, td, *J* = 6.3, 2.3 Hz, H-25), 3.93 (1H, dd, *J* = 7.5, 5.1 Hz, H-14), 5.06 (1H, m, H-20), 5.07 (1H, m, H-16), 5.26 (1H, t, *J* = 6.8 Hz, H-12), 10.24 (1H, s, H-1) (see Appendix A).

#### 3.2.3. Iridobelamal B (**3**)

Colorless oil. HRESIMS [M + Na]^+^ 500.3514 (calcd. for C_31_H_48_O_5_ 500.3502). [α${]}_{D}^{25}$ + 50.4 (c 0.05, EtOH). ^1^H-NMR (500 MHz, Chloroform-*d*): δ 1.3–1.4 (2H, m, H-24), 1.31 (3H, s, H-27), 1.40 (1H, dd, *J* = 6.1, 12.0 Hz, H-10a), 1.62 (3H, s, H-30), 1.69 (3H, s, H-22), 1.71 (1H, m, H-8a), 1.79 (3H, br s, H-3), 1.80 (3H, s, H-28), 1.80–1.86 (1H, m, H-8b), 1.82 (3H, s, H-29), 1.93 (1H, dd, *J* = 8.3, 13.6 Hz, H-10b), 1.98–2.08 (2H, m, H-23), 2.10–2.14 (2H, m, H-19), 2.54 (1H, br d, *J* = 12.0 Hz, H-9a), 2.69 (1H, m, H-9b), 3.39 (3H, s, 26-OMe), 3.61 (2H, td, *J* = 3.0, 6.3 Hz, H-25), 3.66 (1H, br d, *J* = 12.4 Hz, H-5), 4.88 (1H, dd, *J* = 8.2, 16.0 Hz, H-11), 5.11 (1H, m, H-20), 5.11 (1H, s, H-26), 5.48 (1H, br d, *J* = 8.7 Hz, H-12), 5.91 (1H, d, *J* = 10.8 Hz, H-16), 6.16 (1H, d, *J* = 15.3 Hz, H-14), 6.43 (1H, dd, *J* = 10.8, 15.3 Hz, H-15), 10.24 (1H, s, H-1) (see Appendix A).

### 3.3. Inhibitory Effects against Human Neutrophil Elastase

Human neutrophil elastase (EC 3. 4. 21. 37) (Sigma-Aldrich, St. Louis, MO, USA) activity was measured in accordance with the previous description [27] with subtle modification, by observing the formation of *p*-nitroaniline after the hydrolysis of *N*-methoxysuccinyl-Ala-Ala-Pro-Val-*p*-nitro anilide at 405 nm. The inhibitors were dissolved in dimethyl sulfoxide (DMSO) and diluted to a few concentrations. In brief, in a 96-well plate, 10 µL of inhibitor solution and 40 µL of 1.5 mM of MeOSuc-AAPV-*p*NA were added as a substrate in the 0.02 mM Tris-HCl buffer solutions (pH 8.0). Then, 20 µL of human neutrophil elastase (0.2 unit/mL) was added to the mixture. The test mixtures were incubated and mixed for 15 min at room temperature and then screened at 405 nm for 30 min every 30 s. Inhibitory activities were further characterized by determining the concentration required to inhibit 50% of the enzyme activity (IC_50_), which was calculated using the following Equation (1), where [I] is the concentration of inhibitor.


Activity (%) = 100 [1/(1 + ([I]/IC_50_))].(1)


The modalities of HNE inhibition were estimated in experiments using particular concentrations of the substrates and inhibitors, respectively. The Michaelis–Menten constant (*K*_m_) and maximal velocity (*V*_max_) were investigated by a Lineweaver–Burk plot. The *K_I_*, dissociation constants for inhibitor binding to the free enzyme were calculated using a Dixon plot. Equations (2)–(4) are representatives for deriving the aforementioned parameters.
(2)$$\frac{1}{V}=\frac{K_{m}}{V_{\max}}\left(1+\frac{\left[I\right]}{K_{i}}\right)\times \frac{1}{S}+\frac{1}{V_{\max}}$$
(3)$$\mathrm{Slop}=\frac{K_{m}}{K_{i}V_{\max}}\left[I\right]+\frac{K_{m}}{V_{\max}}$$
(4)$$\mathrm{Intercept}=\frac{1}{K_{i}V_{\max}}\left[I\right]+\frac{1}{V_{\max}}.$$

### 3.4. Fluorescence Quenching Measurements

To measure fluorescence from the HNE enzyme, 10 μL of 0.01 unit/mL enzyme solution with 180 μL of Tris-HCl buffer (0.02 mM) was accurately added into the 96-well black immunoplates. Then, 10 μL of incremental concentrations (7.8–31.2 μΜ) of inhibitors were added into each well. All fluorescence spectra were measured from 280 to 400 nm with emission slits regulated to 2.0 nm, and the excitation wavelength was 260 nm. The Stern–Volmer quenching constant (*K*_SV_) was calculated using Equation (5). All experiments were conducted in triplicate, and the mean values were calculated [28].
(5)$$F_{0\;}-{\;F=1+K}_{\mathrm{SV}}\left[Q\right],$$
where *F*_0_ and *F* are the fluorescence intensities in the absence and presence of quencher (Q). *K*_SV_ is the Stern–Volmer quenching constant [LM^−1^]. For static quenching, the correlation between the fluorescence intensity and the concentration of quencher for the series of reactions can be estimated by Equation (6) [29].
(6)$$\log\left[\left(F_{0}-F\right)/F\right]=\log K_{A}+n\log{\left[Q\right]}_{f}.$$

*F*_0_ and *F* are the fluorescence intensities in the absence and presence of inhibitor; *K*_A_ is the binding constant; *n* is the number of binding sites of the enzyme; Q_f_ is the concentration of inhibitor.

### 3.5. Cell Culture and Cell Viability Assay

The mouse macrophage cell line RAW264.7 was obtained from the American Type Culture Collection (ATCC, Manassas, VA, USA). The cells were cultured in DMEM supplemented with 10% fetal bovine serum (FBS), penicillin (100 U/mL), and streptomycin (100 μg/mL) at 37 °C with 5% CO_2_. The medium was replaced every 2 days. After treatment with chemicals, the cell morphological changes were observed under a microscope and images are captured before cell viability assay.

Cell viability was determined calorimetrically using a 3-(4,5-dimethylthiazole-2-yl)-2,5-diphenyl tetrazolium bromide (MTT) reagent (5 mg/mL in phosphate buffered saline (PBS), Duchefa Biochemie, Haarlem, The Netherlands). The MTT assay procedures were performed as described previously [30]. Briefly, a 24-well plate (5 × 10^4^ cells/well) was seeded with RAW264.7, a mouse macrophage cell line. After 24 h of chemical treatment, each well was filled with 20 µL of 5 mg/mL MTT solution (0.1 mg/mL) and incubated for 4 h. After aspirating the supernatants, the formazan crystals in each well were dissolved in 200 µL of dimethyl sulfoxide (DMSO) for 30 min at 37 °C, and the 24-well plates were read at 570 nm with a microplate reader (Bio-Rad, Hercules, CA, USA). Data are expressed as a percentage of viable cells compared with the control.

### 3.6. RNA Isolation and Reverse Transcriptase (RT)-Polymerase Chain Reaction (PCR)

Total RNA was isolated from cultured RAW264.7 cells using TRIzol reagent (Ambion^®^, Carlsbad, CA, USA) according to the manufacturer’s instructions. First strand cDNA was synthesized from the total RNA (3 μg) using oligo dT (DiaStar RT Kit, SolGent, Daejeon, Korea) and was used as a template for PCR amplification with G-*Taq* polymerase (Cosmogenetech, Seoul, Korea). The first-strand cDNA was quantified using a spectrophotometer (NanoDrop^®^ ND-1000, NanoDrop Technologies, Wilmington, DC, USA). The quantified cDNA was used as a template. PCR assay was performed with specific primers for inducible nitric oxide (iNOS), IL-1β, and TNF-α. The primer sequences are listed in Appendix A. The PCR steps included initial denaturation at 94 °C for 5 min, then 28 cycles at 94 °C for 30 s, 57 °C for 30 s, and 72 °C for 30 s, and a final extension step at 72 °C for 10 min. The amplified PCR products were separated in 1.5% agarose gel, and the gel was stained with ethidium bromide. The bands were visualized using the iBright^TM^ CL1500 imaging system (Thermo Scientific Fisher/Life Technologies Holdings Pte Ltd., Singapore). The DNA fragments were directly sequenced with the ABI PRISM^®^ 3100-Avant Genetic Analyzer (Applied Biosystems, Carlsbad, CA, USA).

### 3.7. Measurement of Nitrite Levels

Nitrite levels were determined using the Griess method, as described previously (Jeong et al., 2016). Briefly, RAW264.7 cells (5 × 10^4^ cells/well) were cultured in a 24-well plate. The cells were pretreated with triterpenoids (1 or 3 µg/mL) 2 h prior to lipopolysaccharides (LPS, 1 µg/mL) treatment for 16 h. Supernatants were collected and centrifuged at 2451× *g* for 10 min. The (50 μL) of supernatants were mixed with equal volumes of Griess reagent (1% sulfanilamide, 0.1% *n*-l-naphthylethy-lenediamine dihydrochloride, and 2% phosphoric acid), and incubated for 10 min. The optical density was measured at 550 nm, and the nitrite levels were calculated from a standard curve generated from sodium nitrite.

### 3.8. Measurement of Cytokines Concentration

RAW264.7 cell (2 × 10^5^ cells/well) were cultured in a 6-well plate for 24 h and treated with compounds (1 or 3 µg/mL) 2 h prior to LPS treatment, followed by LPS (1 µg/mL) for 16 h. The concentration of pro-inflammatory cytokines in the collected supernatant was quantified using an ELISA kit (R&D system, Minneapolis, MN, USA) according to the manufacturer’s instructions. Briefly, 50 μL of assay diluent, sample, and standards were added to the ELISA well plate, which were pre-coated with anti-mouse IL-1β and TNF-α antibody. The plates were then covered with adhesive strip, agitated, and incubated for 2 h at room temperature. The ELISA well-plate was washed five times with a wash buffer, then 100 μL of mouse IL-1β and TNF-α conjugates was added and incubated at room temperature for 2 h. The plate was washed five times, 100 μL of substrate solution was added to each well, and incubated for 30 min in the dark. The reaction was quenched by the addition of 100 μL stop solution to each well, and the absorbance was read 450 nm/570 nm with an ELISA reader (Molecular Devices, Sunnyvale, CA, USA).

### 3.9. Immunocytochemistry

RAW264.7 cells cultured on round cover slips coated with poly-L-lysine were treated with triterpenoids (1 or 3 µg/mL) 2 h prior to LPS treatment, followed by LPS (1 µg/mL) for 16 h. The cells were washed with 1 x phosphate buffered saline (PBS) and fixed with 4% paraformaldehyde for 30 min. After washing the fixed cells with PBS, the cells were permeabilized with a blocking buffer (1% normal goat serum and 0.2% Triton X-100 in PBS) for 30 min at room temperature, and were incubated with an anti-NF-κB p65 antibody (1:200 dilution, Cell Signaling Technology, Danvers, MA, USA) at 4 °C overnight. After three washes in PBS, the cells were incubated in the dark for 1 h with the mixture of FITC-conjugated anti-rabbit IgG fluorescent secondary antibody diluted at 1:500 in PBS. Finally, the cells were stained with 2 µg/mL Hoechst for nuclei staining. The cells were washed with PBS, wet-mounted on glass slides, and observed using a confocal laser-scanning microscope (Olympus, Tokyo, Japan). The negative control (NC) was analyzed by omitting the primary antibody.

### 3.10. Western Blot Analysis with Nuclear and Cytoplasmic Fractions

RAW264.7 cells were homogenized in a hypotonic lysis buffer and centrifuged at 142× g for 5 min at 4 °C. The resulting pellets and supernatants were used for the isolation of nuclear and cytosoplasmic fraction, respectively. The pellets were incubated on ice for 5 min with a nuclear isolation buffer and were spun at 142× *g* using Eppendorf centrifuge 5424 R at 4 °C for 5 min. The resulting pellets were incubated on ice for 30 min with 2% Triton nuclear isolation buffer and were spun at 12,448× *g* at 4 °C for 15 min. For the cytosolic fraction, the resulting supernatants in the hypotonic lysis buffer were transferred to a new 1.5 mL tube and spun at 15,401× *g* for 15 min at 4 °C. The resulting supernatants were centrifuged at 100,000× *g* at 4 °C for 60 min in an ultracentrifuge (TLA 100.3; Optima MAX-XP, Beckman Coulter, Inc., Brea, CA, USA). The proteins of nuclear and cytosolic fractions were separated on 8% SDS-PAGE gel and blotted onto polyvinylidene difluoride membranes. Equal amounts of 35 µg were analyzed by Western blotting. Briefly, the membranes were blocked with 5% fat-free dry milk and then incubated with the anti-NF-κB p65 polyclonal antibody (1:1000 dilution; Cell Signaling Technology), and anti-β-actin monoclonal antibodies (1:5000 dilution). The primary antibody incubation was followed by incubation with a secondary horseradish peroxidase-conjugated goat anti-rabbit or goat anti-mouse antibody at 1:5000 dilution (Assay Designs, Ann Arbor, MI, USA). Immunopositive bands were visualized by iBright™ CL1500 Imaging System (Thermo Scientific Fisher/Life Technologies Holdings Pte Ltd., Waltham, MA, USA). The cytosolic and nuclear fractions were verified to be free of contaminating nucleus and cytosol by immunoblotting for the cytosolic marker β-actin and the nuclear marker lamin, respectively.

### 3.11. Statistical Analysis

One-way ANOVA with post hoc comparisons using the Bonferroni test were used to analyze the difference among groups (OriginPro2020, OriginLab Corp. Northampton, MA, USA). Data were presented as the mean ± S.D., and significance was set at *p* < 0.05.

## 4. Conclusions

*B. chinensis* has been considered a representative medicinal plant for inflammatory diseases such as bronchitis, asthma, and sore throat. Iridal-type triterpenoids (**1**–**3**) were found to effectively inhibit the pro-inflammatory enzyme, human neutrophil elastase (HNE). The Stern–Volmer quenching constant (*K*_sv_) and the binding constant (*K*_A_) were in accordance with inhibitory potencies (IC_50_). The HNE enzyme had only a single binding site for iridal-type triterpenoids. The spiro-type compound **3** showed a **2**–**3** fold better HNE inhibition and binding affinities than the monocyclic ones (**1** and **2**). Among them, isoiridogermanal (**1)** and iridobelamal A (**2**) effectively suppressed the expression of pro-inflammatory cytokines, such as iNOS, IL-1β and TNF-α through blocking the NF-κB pathway. In summary, iridal-type triterpenoids might be prime phytochemicals for the anti-inflammation potential of *B. chinensis*.

## Figures and Tables

**Figure 1 molecules-26-06602-f001:**
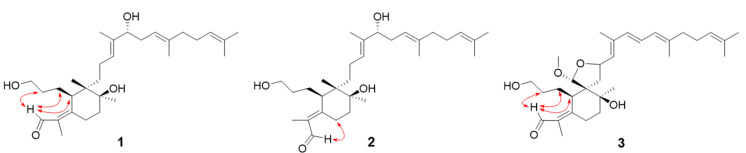
Structures of iridal-type triterpenoids from the roots of *B. chinensis* and their key NOESY correlations.

**Figure 2 molecules-26-06602-f002:**
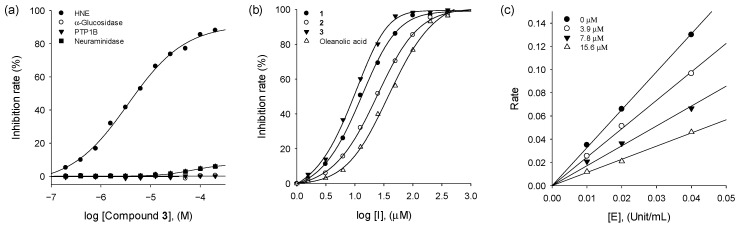
(**a**) Selectivity of compound **3** on HNE in different hydrolase enzymes. (**b**) Dose-dependent HNE inhibition by iridal-type triterpenoids (**1**–**3**). (**c**) Determination of the reversible inhibitory mechanism of **3**.

**Figure 3 molecules-26-06602-f003:**
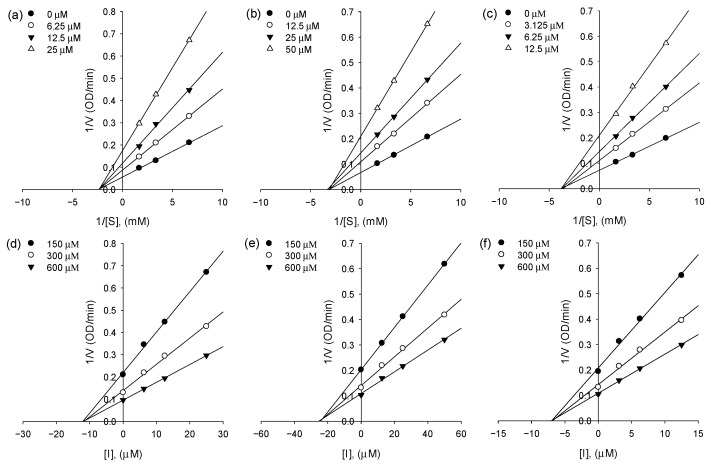
Kinetics and binding affinities of isolated iridal-type triterpenoids on HNE. (**a**–**c**) Lineweaver–Burk plots for the effect of **1**, **2**, and **3** on the HNE. (**d**–**f**) Dixon plots for the effect of **1**, **2**, and **3** on the HNE.

**Figure 4 molecules-26-06602-f004:**
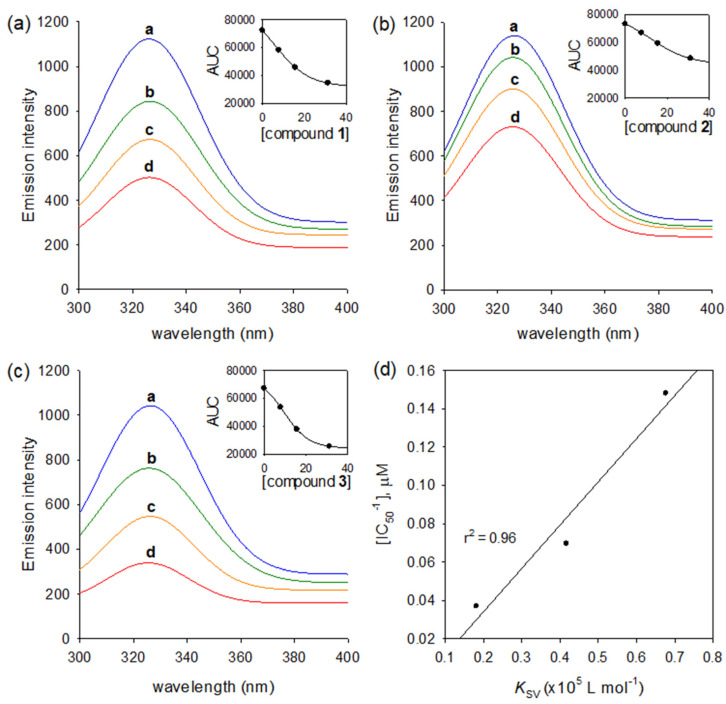
Binding affinities of isolated iridal-type triterpenoids on HNE. (**a**–**c**) The fluorescence emission spectra of HNE at different concentrations of compound **1**, **2**, and **3** (0, 7.8, 15.6, and 31.2 μM for curves from a to d). (**d**) The correlation between half maximal inhibitory concentration (IC_50_) values and Stern-Volmer constants (*K*sv) of compounds **1**–**3**.

**Figure 5 molecules-26-06602-f005:**
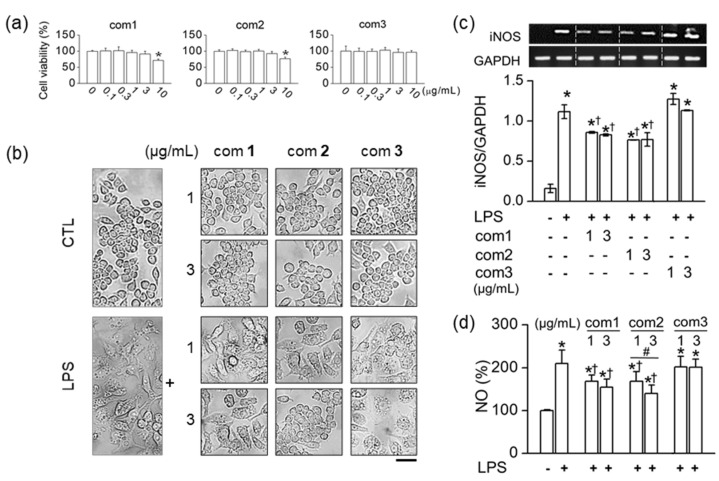
Effects of iridal-type triterpenoids on LPS-stimulated RAW264.7 cells. (**a**) Effect of the triterpe-noids at different concentrations on the cell viability. (**b**) Inhibition of LPS-induced macrophage activation by pre-treatment with the triterpenoids. Scale bar, 50 μm. (**c**) Inhibition of LPS-induced upregulation of iNOS mRNA expression by the triterpenoids. (**d**) Inhibition of LPS-induced increase in NO production by the triterpenoids. (**a**,**c**,**d**) * *p* < 0.05 compared to control (no treatment with LPS); ^†^
*p* < 0.05 compared to LPS alone treatment; ^#^
*p* < 0.05 compared to the 1 μg/mL of compound **2**.

**Figure 6 molecules-26-06602-f006:**
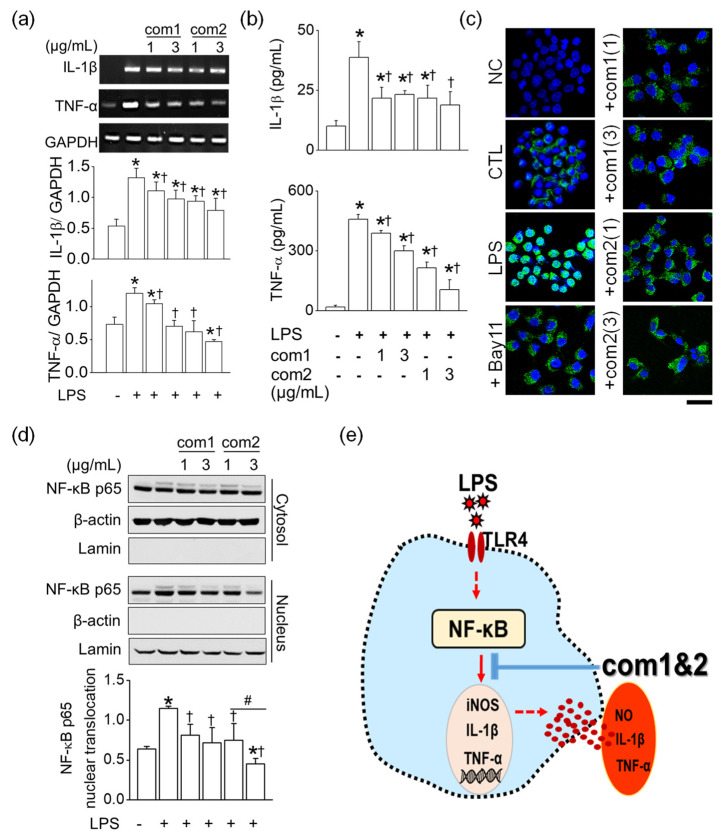
Suppression of pro-inflammatory cytokines through blocking NF-κB pathway by iridal-type triterpenoids in LPS-stimulated RAW264.7 cells. (**a**,**b**) Down-regulation of IL-1β and TNF-α mRNA expression and concentration by triterpenoids. (**c**,**d**) Suppression of LPS-induced nu-clear translocation of NF-κB p65 upon treatment with triterpenoids confirmed by immunocyto-chemistry and Western blotting assay, respectively. NC represents negative control without primary antibody treatment. CTL, LPS, and Bay 11 represent control, lipopolysaccaride, and Bay11-7085, respectively. Bay11-7085 was used as an inhibitor of NF-κB p65 translocation. Scale bar, 20 μm. The plus and minus signs (+ and −) indicate conditions with and without treatment, respectively. (**a**,**b**,**d**) * *p* < 0.05 compared to control (no treatment with LPS); ^†^
*p* < 0.05 com-pared to LPS alone treatment; ^#^
*p* < 0.05 compared to the 1 μg/mL of compound **2**. (**e**) Schematic representation of anti-inflammatory signals in LPS-stimulated RAW264.7 cells by triterpenoids.

**Table 1 molecules-26-06602-t001:** Inhibitory effects of iridal-type triterpenoids (**1**–**3**) on HNE activity.

Compounds	IC_50_ ^a^ (µM)	Type of Inhibition (*K*_i_ ^b^, µM)
**1**	14.4 ± 0.3	Noncompetitive (12.7 ± 0.3)
**2**	27.0 ± 0.6	Noncompetitive (24.9 ± 0.5)
**3**	6.8 ± 0.3	Noncompetitive (6.2 ± 0.3)
Oleanolic acid ^d^	38.8 ± 0.8	NT ^c^

All compounds are tested in three sets of experiments; ^a^ IC_50_ values of compounds represent the concentration that caused 50% enzyme activity loss; ^b^ Values of inhibition constant; ^c^ NT is not tested; ^d^ Oleanolic acid is a positive control.

**Table 2 molecules-26-06602-t002:** Evaluation of Stern–Volmer constants regarding fluorescence quenching effects of HNE inhibitors **1**–**3**.

Compounds	*K*_SV_^a^ (×10^5^ L mol^−1^)	*K*_A_^b^ (×10^6^ L mol^−1^)	*n* ^c^
**1**	0.4166	0.0411	1.0039
**2**	0.1816	0.0199	0.9734
**3**	0.6764	0.0743	0.9727

^a^*K*_sv_ value of each compound represents the Stern-Volmer quenching constant; ^b^ Association constant of the complex of enzyme and quencher; ^c^ The number of binding sites of the enzyme.

## Data Availability

The data presented in this study are available on request from the corresponding author.

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
