# Peer review of "Iridal-Type Triterpenoids Displaying Human Neutrophil Elastase Inhibition and Anti-Inflammatory Effects from Belamcanda chinensis"

_molecules, 2021, doi:10.3390/molecules26216602_

Round 1
Reviewer 1 Report
In the manuscript, four iridal-type triterpenoids were isolated from Belamcanda chinensis and identified as isoiridogermanal, iridobelamal A, iridotectoral C, and iridobelamal B. And, their human neutrophil elastase 2 inhibition and anti-inflammation effects were evaluated and investigated that compounds 1-4 exert potent anti-inflammatory activity in two different ways. First, they examined the inhibitory activity and also specificity of compounds 1-4 on HNE and also evaluated the type of enzyme inhibition by biochemistry analysis in tube assay. Second, authors also evaluated and investigated the anti-inflammatory effect of 1-4 in LPS-stimulated RAW 264.7, a well-established in vitro cell model. Overall, the bioactivity study of the compounds is convincing but there are some inappropriate formats or description could be improved.
- In the manuscript, there is a serious mistake that one of the isolated compounds is incorrect. The NMR spectra (Supporting information) of compound 3 (Iridotectoral D) were different with the reported literature ( Pharm. Bull. 2002, 25, 432-436). Please check carefully and revised.
- Line 94. – “27.0 μM.” 27 is not necessary with bold type.
- Line 114-116 (Table 1.) –The font size inside the table should be uniform. 3 and 4 showed different font size to 1 and 2. In addition, the notation of IC50 also should be consistent: IC50 in footnote but IC50 in table.
- Line 152. – “The Ksv values were ranked in following order 4 > 3 > 1 > 2”.
- The Ksv values were calculated according the results of fluorescence quenching assay of HNE (Figure 4a-4c). However, Figure 4. Only showed the results of 1, 2 and 4, but not of 3 (although I found that it has actually been provided in supplementary data). Authors should add a description to notice that.
- Line 182-183. – “Among the four triterpenoids, 1 and 2 significantly ameliorated the LPS-induced morphological changes in RAW264.7 cells.” Although 2 showed effect on suppression of LPS-induced morphological change, I couldn’t see the differences between 1 and 3, 4 on morphology. Authors may replace the representative images with the clearer ones.
- For Figure 5 – Figure 5b: The morphological images are not clear, the resolution is too bad. Authors should provide images with better resolution, at least clear to see the morphological changes such as cell enlargement, flattening, cellular spreading, and vacuole formation. Figure 5c: (1) Please provide the quantification data of the mRNA analysis with statistical analysis. (2) The bar chart of NO production (%) should be separated to Figure 5d. The combination of iNOS mRNA analysis and the NO production (%) is easily to confuse and to ignore that they are two results of different assay.
- For Figure 6 – Please provide the quantification data of the mRNA analysis with statistical analysis. Figure 6c: The abbreviations of each treatment should be explained (like what the NC stands for) in figure legend as well as scale bar size. Figure 6d: The y-axis tile of quantification data, “NFkB” nuclear translocation is not appropriate. NFkB is consisted of p50 and p65 but the nuclear translocation is only occurred for p65 not for p50. Therefore, “p65 nuclear translocation” is better.
- In supplementary material – there are many errors, such as (1) “ S1. 1H-NMR spectrum of compound 1 (500mHz, Acetone-d6)” should be revised as “Fig. S1. 1H-NMR spectrum of compound 1 (500 MHz, chroloform-d4)”. (2) “Fig. S2. 13C-NMR spectrum of compound 1 (500mHz, Acetone-d6)” should be revised as “Fig. S1. 13C-NMR spectrum of compound 1 (125 MHz, chroloform-d4)”. (3) The scale range of DEPT should be 0-200 ppm because compound 1 possess the aldehyde group. (4) “Fig. S1. DEPT-90 and -135 spectrum of compound 1” should be revised as “Fig. S1. DEPT-90 and -135 spectra of compound 1”. (3) The 1H-NMR scale range of HMBC should be 0-10 ppm. Please revised the supplementary materirial of the others according to the above suggestions.
Reviewer 2 Report
The manuscript "Iridal-type triterpenoids displaying human neutrophil elastase inhibition and anti-inflammation effects from Belamcanda chinensis", deals with the antiinflammatory potential con triterpenoid compounds isolated from B. chinensis.
The manuscript is interesting and I consider that the assays were performed properly.
I recommend reviewing English along the text, since some sentences are not adequately written.
Some minor corrections and suggestions are indicated in the pdf file

Round 2
Reviewer 1 Report
Please see the attached file.

Round 3
Reviewer 1 Report
As attached.
